# Personal Tools and Psychosocial Resources of Resilient Gender-Based Violence Women

**DOI:** 10.3390/ijerph18168306

**Published:** 2021-08-05

**Authors:** Rebeca García Montes, Inmaculada Corral Liria, Raquel Jimenez Fernandez, Rocío Rodriguez Vázquez, Ricardo Becerro de Bengoa Vallejo, Marta Losa Iglesias

**Affiliations:** 1Nursing and Stomatology Department, Health Sciences Faculty, King Juan Carlos University of Madrid, 28922 Madrid, Spain; r.garciamon@alumnos.urjc.es (R.G.M.); raquel.jimenez@urjc.es (R.J.F.); rocio.vazquez@urjc.es (R.R.V.); marta.losa@urjc.es (M.L.I.); 2Nursing Department, Nursing, Physiotherapy and Podiatry Faculty, Complutense University of Madrid, 28040 Madrid, Spain; ribebeva@enf.ucm.es

**Keywords:** gender-based violence, coping, abuse, survival, resilient

## Abstract

Gender-based violence is considered a serious social and public health problem. Overcoming this situation implies a process that results in the favorable biopsychosocial rehabilitation of the resilience of women. The objective of this study was to analyze the tools, resources and personal and psychosocial mechanisms used by women survivors of gender-based violence. The design was an interpretative phenomenology. It was carried out with 22 women who have overcome gender-based violence. Data were collected through personal interviews and narration. The results were grouped into four themes: “Process of violence”, “Social resources for coping and overcoming GBV”, “Personal tools for coping and overcoming GBV” and “Feelings identified, from the abuse stage to the survival stage”. Several studies concluded that overcoming abuse is influenced by the women’s social network, and it can be the action of these people determining their survival to gender violence. Despite the recognized usefulness of these available resources, it would be desirable to strengthen them in order to be able to drive more women toward survival, assuming a strengthening of coping and overcoming, without forgetting the importance of other support mechanisms, such as their family and group therapies.

## 1. Introduction

Gender-based violence (GBV) is one of the most important social problems in Spain, given both the increase of abuse and a much higher number of cases in 2020 [1]. It is defined as any act of physical or psychological violence, including aggressions against sexual freedom, threats, coercion or arbitrary deprivation of liberty, both in public or private settings, carried out by a current or past spouse of the woman, or by anyone who is or has been related to her through a similar personal relationship, even if they do not live together [2].

The Macro Survey on Violence Against Women, conducted by the Spanish Ministry of Health, Social Services and Equality (2019), found that over 11% of Spanish women have suffered some type of physical aggression by a current or past partner. These were repeated aggressions in 75.2% of the cases, and single incidents in slightly over 12%. These figures are quite alarming, as they indicate that in most cases of GBV, aggressions are repeated over time. In 49% of the cases, the aggressions were by former partners, while less than 5% were committed by current partners [3].

When addressing women’s perception regarding the consequences of GBV, the severity increases when the perpetrator is an ex-partner (46.2%), while it drops by more than half if it is the current partner (24.5%). After an attack, women most frequently feel powerless and sad [3].

In 2019, 85,318 calls were received at the National Information and Legal Advice Hotline for women victims of GBV across the country, while 143,535 police reports were filed. In the province of Huelva, 925 calls were received, with over 60% of them coming from the women themselves, and 1,932 police reports were filed. In addition, one woman died in the province, the first since 2011 [1].

Overcoming GBV is a biopsychosocial rehabilitation process for women and their children, during which post-traumatic growth takes place to devictimize them and separate them from any negative consequences, such as mental illness. The process includes enhancing the natural ability of human beings to resist and overcome adversities [4]. This is known as resilience and in order to achieve it GBV must be confronted. This term was defined by Lazarus and Folkman [5,6,7] as “those constantly changing cognitive and behavioural efforts to manage specific internal and/or external stressors that are appraised as taxing or exceeding the resources of the person”. These same authors differentiated two coping functions: one aimed at the problem, which includes the person and his or her efforts to deal with it, and the other aimed at the emotion, which attempts to minimize the emotional impact of the conflict and may include avoidance or minimizing the problem or staying away from the aggressor

Throughout the episodes of violence, the influence of the women’s social network should be taken into account, primarily the natural one made up of their family members and/or trusted friends. When these people learn about the situation, they can react with feelings of rage or anger, consequently victimizing the aggressors, or they can create conflicts between the women and these individuals, given their emotional dependence on them. Other family members may blame the victims for the violence or even feel ashamed of the situation. They may try to control them or even play down the abuse, doubting the credibility of the women’s testimony [8,9,10].

From an institutional perspective, the action of an organized social network, made up of government and non-government resources [11], which are very diverse in Spain, may be crucial during the process [1].

### Aims

The objective of this study was to analyze the tools, resources and personal and psychosocial mechanisms used by women survivors of gender-based violence.

## 2. Materials and Methods

### 2.1. Design

This is a qualitative study using interpretative phenomenological analysis, in line with Heidegger’s principles [12,13,14,15].

### 2.2. Participants

The sampling used was theoretical and purposeful [16,17], with the participation of women who had suffered GBV and had been able to overcome it. For the sample, we contacted the association Miriadas in Huelva, created in 2002 and devoted to women victims and their minor children. The association provides an emergency hotline operating 24 hours a day, accompaniment and follow-up during the process of reporting the aggression to the police, assistance when abandoning the situation of abuse and weekly training and mutual support workshops [18]. The final study sample was 22 women, which achieved data saturation (Table 1). The data obtained were collected between January 2019 and January 2020.

The inclusion criteria considered were women of legal age (over 18), residents in Spain, separated from their aggressors with whom they had had no contact for 3 years or more, active in the labor market (either working and/or looking for a job) and who voluntarily decided to participate in the study.

The exclusion criteria were women for whom talking about their experience of abuse caused physical and/or mental suffering and/or women living in short and/or medium stay social institutions such as shelter homes.

### 2.3. Data Collection

Data were collected through semi-structured interviews carried out during personal appointments with 4 of the participants, while the rest decided to provide their data in written semi-structured reports with the same questions used in the interviews (Table 2).

### 2.4. Data Analysis

The data analysis strategy followed Colaizzi’s approach [19,20], which is divided into seven points:We read the informants’ description in order to acquire and compare contents;We obtained significant statements for each description and coding;We explained the meaning in detail of each significant statement in order to formulate meanings and continue coding;We organized the complete set of statements was into groups;We compared the groups with the original comparisons in order to validate the groups and find discrepancies;We conducted an exhaustive description of the phenomenon, integrating the results from the previous steps;We consulted participants in order to validate the original data.

The study was subject to the four principles of bioethics. The first three were the principle of beneficence, which aimed to use the data collected in the interviews to help women who are currently victims of abuse, while respecting the participants in the study, the principle of non-maleficence, under which no woman may be intentionally harmed and the principle of autonomy, which was accepting and respecting the decisions of the women, who voluntarily participated in the study and could leave it at any time. To this end, they signed an informed consent (IC). The fourth principle was the principle of justice, which guided the study, as the intention was to discover the most appropriate mechanisms used by the women who had been through this type of situation and make them available to those who needed them [21,22].

All subjects gave their informed consent for inclusion before they participated in the study. The study was conducted in accordance with the Declaration of Helsinki, and the protocol was approved by the Ethics Committee of 1801201702017.

The rigor of this study was determined by the following criteria: (a) confirmability is guaranteed by describing the women’s characteristics and by using a literal transcription of their input; (b) transferability is achieved since the results obtained can be transferred to other similar studies [23,24,25].

## 3. Results

Results were grouped into four themes: “Process of violence”, “Social resources for coping and overcoming GBV”, “Personal tools for coping and overcoming GBV” and “Feelings identified, from the abuse stage to the survival stage” (Table 3).

### 3.1. Process of Violence

The process of violence included a series of aggressions, a process of identification, an initial action by the women to stop the abuse and a series of consequences.

#### 3.1.1. Types of Aggressions Suffered by the Women

The aggressions the women reported were physical. They were objective injuries that could trigger a reaction from family members or social workers as well as causing a mark for life that they tried to conceal, while sexual aggressions went unnoticed.

-
*“Because of course we couldn’t have sex with a condom, right? Then it happened, aside from, well, he didn’t force me to have sex, but it was a bit like we had to do it.” (16)*
-
*“He beat me up terribly and destroyed my house.” (7)*


Additionally, there was psychological harm that intimidated them and made self-defense impossible. The aggressor’s constant manipulation made them feel like they had no value as a person and it socially destroyed their natural support network by distancing them from the most important people in their lives, which made them feel alone and abandoned.

-
*“He would say that control was love, but I knew it was possession.” (13)*
-
*“He would yell at me for everything and make me feel I was worthless.” (2)*
-
*“I started to feel dirty, alone and despised.” (7)*


Similarly, a series of financial aggressions were identified. They were stripped of their property rights, belongings that had monetary and sentimental value were destroyed and they felt vulnerable and unable to make decisions.

-
*“He wouldn’t give me money for the house, he would go to play bingo and when he got home he would beat me, and force me to do things I didn’t want to do.” (12)*


#### 3.1.2. Identification of Violence and Actions to Stop It

During the identification process, women refused to see themselves as victims. They did not accept the image imposed on them by society of being weak, blinded by love, of low social class and with no education, which was a hard blow for their self-esteem. Furthermore, some of them did not think that their partners were abusing them until the aggressions became physical and they became afraid, because the psychological aggressions they had experienced were not considered acts of gender violence.

-
*“Well, let’s see. For a long time I did not want to feel like a victim. So it was like, I am going through everything that could objectively be called gender abuse, but I am not an abused woman, but rather this is something that happened to me.” (16)*
-
*“When I truly began to fear for my physical integrity I told him I wanted to leave him, I’d had enough.” (16)*


This process of identification would include their children’s suffering, an attempt to get out of the situation, although unsuccessfully, while maintaining a bond with the aggressor and returning to the acts of aggression that got even worse.

-
*“He started abusing his daughters, not only me.” (8)*
-
*“The trigger was my daughters’ suffering.” (1)*


After all these experiences, and in an attempt to improve the situation, the women would begin to take steps, such as reporting the violence to the police, breaking up the relationship and/or leaving home and calling for help.

-
*“To stop the abuse I reported it to the police and went to a shelter.” (5)*
-
*“I left him the moment I knew it was abuse.” (2)*


#### 3.1.3. Consequences of Abuse

The results or consequences were devastating, including the onset of non-specific illnesses, sleep disorders, anxiety, nightmares, emotional stress, hypersensitivity towards social image and the inability to see the future. Some of them considered that they had not yet overcome the violence since their memory of the trauma was visible in their everyday life.

-
*“Yes, I have physical sequelae. He gave me the human papillomavirus.” (16)*
-
*“Many physical sequelae and even more psychological ones”. (9)*
-
*“My body reacts by sweating a lot, not sleeping much, with diarrhea, vomiting and panic attacks, anxiety.” (9)*
-
*“I have nightmares about him. There is a nightmare that I had several times where I am in different situations and I see his head at a distance and I feel horribly terrified” (16)*


### 3.2. Social Resources for Coping and Overcoming GBV

#### 3.2.1. Natural Social Support Network

Access to resources available for coping and overcoming violence was different in each case. Some had a natural social support network made up, primarily, of family members and friends, who would provide mental and physical support that could favor their empowerment, detachment from their aggressor and make them feel protected. In other cases, the women did not have anyone nearby or people nearby offered to give them a hand.

-
*“I received support and affection from all my loved ones.” (7)*
-
*“My family distanced themselves from the situation, they stayed away.” (6)*
-
*“My parents were the best and greatest support I had, and with my siblings I have no relationship because they say I am to blame for everything that I have gone through.” (9)*


On the hard path towards resilience, some decided not to talk about their experience to avoid worrying those around them, to not appear weak or ashamed and also to prevent confrontations with the people they trusted.

-
*“I never told my family, due to shame and because I didn’t want to cause any more problems at home.” (13)*


Additionally, they may have come to distrust their support network because nobody believed there was abuse and defended the aggressor. Likewise, they may have felt like they were being pressured into leaving the aggressor, since the others did not understand their emotional dependence on the man or their need for some time to think things through.

-
*“My family’s attitude was very bad, they never helped me. When I was at the first shelter he was living at my mother’s house.” (5)*
-
*“And then I had a friend who would tell me off, like: “you go back with him, you are messing up the people around you....” (16)*


This type of behavior by the people closest to the women were assessed as a wish for greater support or as gratitude; however, at times support was rejected since they felt that they had to overcome the violence alone.

-
*“I would have liked help from my family, phone calls to ask how I was, showing interest in me and my children.” (5)*
-
*“I would have changed getting more support from them, not to be left alone.” (6)*
-
*“I took this as something that was mine ‘This is mine, I’m going to deal with it and I’m the one asking for help’ and I went to talk to the psychologist all by myself.” (16)*


#### 3.2.2. Organized Social Support Network

With or without this natural network, they can also ask for help from organized support networks, such as the Association Miriadas, psychology professionals, therapy groups or the Andalusian Women’s Institute (IAM) and shelters. These organizations can help them feel accompanied, make their feelings of shame disappear and provide an environment where they do not feel judged and where they can receive comprehensive and personalized care.

-
*“I learnt about Miriadas where I have been able to grow as a person and heal the consequences of the abuse. It is made up of women who were all victims of violence, and its director was also a victim. She helps us and understands us. She experienced this problem in her own flesh. She is a wonderful person and I will never forget her, she made me grow as a person and heal my wounds.” (8)*
-
*“Mutual help workshops provided by Miriadas.” (9).*


At times, they may feel dissatisfied with the available resources, since they demand more social assistance to facilitate their inclusion, which may not be sufficient to meet the needs of all women.

-
*“The lack of sensibility they show us in court.” (6)*
-
*“I miss more financial resources and justice.” (1)*
-
*“There is no help for victims, and most importantly, for their children, also psychological help.” (8)*
-
*“We need assistance for a place to live, the waiting lists are very long.” (7)*
-
*“I felt helpless, without a home and with children.” (10)*


### 3.3. Personal Tools for Coping and Overcoming GBV

During the process, they improved their social skills, felt capable of accepting and facing all the adversity that they were facing and reinforced their self-esteem. All of these tools were in addition to other strategies they had been used previously, such as support and testimonials from other women victims of abuse or doing leisure activities that had been prohibited during the period of violence and that they valued more now. All of them proved they were independent from their aggressors and that made them feel free.

-
*“We established a new circle of friends, I started mixing more.” (1)*
-
*“Now I face whatever comes, even if I lock up at first, I keep going.” (9)*
-
*“Being able to express how I feel without fear of being misunderstood, of feeling inferior.” (6)*
-
*“Once I came out of it all I realized I could manage on my own and I no longer had to depend on him as I had thought.” (4)*


### 3.4. Feelings Identified, from the Abuse Stage to the Survival Stage

#### 3.4.1. When My Life Was All Darkness

In contrast, throughout the process, they would have feelings such as disgust for the situation of abuse and even for themselves given their insecurity and low self-esteem, guilt, shame, fear, anxiety, rage, loneliness and negation of self-worth. In general, they were ambivalent, negative or unspecific, because they felt like they did not matter to anyone and that nobody except their aggressors paid attention to them.

-
*“He made me feel like I was worthless.” (2)*
-
*“I felt guilty, dirty, insignificant.” (9)*
-
*“The feeling I get when I remember it is sadness, sadness for having allowed it and for not having ended it all sooner.” (13)*
-
*“I feel very angry for having allowed this to happen, you know? Because I swear I realized something wasn’t right from day one.” (16)*
-
*“Yes, of course. I felt like a piece of shit because I didn’t want to break up. I felt really confused.” (16)*


#### 3.4.2. My World Is Beginning to Shine

The women described feelings of calm, relief, indifference, struggle, independence, happiness, strength and a sense of overcoming when time had passed and they had survived. Once they had overcome abuse, their memories of that period gave them a feeling of satisfaction and well-being, a feeling of pride of having made it after a long and difficult *struggle.*

-
*“I am happy and I feel inner peace, and I am proud of being a survivor of this social blight.”(8)*
-
*“I felt relieved and strong enough to give myself a chance without him.” (13)*
-
*“I know I am strong and can be independent and self-sufficient, I keep telling myself this over and over again.” (13).*


## 4. Discussion

The study conducted by Crane et al. [26] coincides with ours by treating the type of violence suffered by women, which is mainly physical and psychological aggression. In that paper, sexual violence was acknowledged by only 20%, which could be due to the fact that they did not consider it to be important, or they included it within physical violence, as was the case in our research. Likewise, the study by Lutwak [27] considered that the symptoms of post-traumatic stress disorder, such as avoidance of memories, intrusive dreams or the feeling of a hopeless future, are characteristic of women victims of GBV. Ruxana et al. [28] combined them all in the battered woman syndrome, which is in line with the testimonies of the women in our study, who identified these symptoms as consequences of the traumatic situation experienced.

Thus, Morales et al. [29] cited some characteristics of individual resilience, such as self-esteem, independence or relational capacity, all mentioned by the women in our study as personal coping tools. We confirmed that the women in our research showed that they were able to move on.

Neetu [30] identified several factors that had an impact on both the psychological effects and the women’s efforts to get out of the abusive situation. For example, the response of the OSN (online social network) may have conditioned these effects, leading to a second victimization, or the women’s material resources and their NSN (natural social network) may have helped them to decide to end the violence either sooner or later. This matches what the women in our study reported about their social network. The actions taken by the resources to whom they turned for help were very important, since they confirmed their suspicion of gender-based violence and provided guidance. However, if these factors were not taken into account, the women did not feel the support they were looking for and ended up reaffirming themselves as victims that could not get away. This is what the women in our study demanded, since they considered that they had been attended to following a standard operating procedure without asking about their unique circumstances or offering them the resources that best matched their needs.

Ogbe et al. [31] showed the relationship between social support and gender-based violence and how the possibility of being abused diminishes by more than half if this support is taken into account. Therefore, support of the NSN as a factor for avoidance of violence can also be considered, which does not mean that victims cannot get out of abuse even if this is missing. This issue was likewise confirmed by the women in our study.

Bucuţă et al. [32] conducted research in collaboration with the Spanish Women’s Institute on the emotions of women during the process of overcoming GBV, from the start of the abuse up to resilience. She distinguished between two groups of feelings and compared them to the intercultural social roles of women: the blocking feelings, which were those that would manifest when there was an emotional dependence that prevented women from detaching themselves from the aggressor, with feelings of guilt and shame, and the facilitating feelings that enable empowerment and dissociation from the aggressor, when women feel stronger and express a willingness to fight. The former were represented in our study during the stage of explicit violence, where the women’s emotional dependence and vulnerability prevented them from identifying the abuse. This changed as the relationship progressed: they acknowledged that they were victims of GBV and began to take on the facilitating feelings, which encouraged them to react against the relationship. Crann et al. [33] talk about multiple cognitive, emotional and behavioral shifts in women, controlling themselves, resisting at certain times and being positive in all possible situations.

The study conducted by Brandt et al. [34] addressed the reasons why women would withdraw the charges that they had made to the police and the meaning of this. This author stated that the charges were seen by society as the victim’s attempt to get out of the abuse, when in reality it is one of the various possible paths to leave violence behind. This socially imposed idea is strongly linked to media campaigns, which have generated expectations about an immediate resolution if you go to the police. Since this is not what women actually experienced: it made them feel disappointed and defeated once again, because they had made another effort, with no results. This idea was expressed by some participants in our study who felt neglected by the judicial system or were warned of that by OSN resources before taking action to stop the violence.

### 4.1. Limitations

The search for participants was very difficult, since women who have suffered GBV have a hard time talking about it, and it is almost impossible to contact them directly, having to resort to the mediation of associations.

Participants reported that it would have been better to include their children in the study as they consider them an inseparable part in GBV, but this consideration was not included in the objectives of the research.

### 4.2. Lines of Further Research

To study the consequences of gender-based violence and overcoming it on the descendants of the surviving women.

To analyze the feelings and reflections of aggressors with regard to the abuse once their sentence has been completed, to improve the existing resources against gender-based violence and to create additional new ones.

## 5. Conclusions

The personal tools for coping with and overcoming GBV most frequently used by the women in this study were those involving third parties: either people from their natural social network or other survivors. On the path toward resilience, many discovered other tools, such as independence or increased self-esteem.

The women who had suffered GBV knew about and used the psychosocial resources available in order to overcome the traumatic experience, but pointed out the need to increase and improve them. They considered those that are provided by organized social support networks to be very useful, although some women disagreed with their actions. The natural social networks, which have an important influence on women, offered everything from strong support for the woman to no support at all, or they even sided with the aggressor. Women were grateful for the behavior of these people, but in general demanded more support.

The feelings reported by the survivors of GBV were mostly negative towards themselves and the situation that they were living in, turning against the aggressor once GBV had been overcome, which is when the feelings of independence, survival, strength and a willing to fight emerged.

In summary, the psychosocial resources available have been very useful for the women who wanted to get out of GBV, although reinforcing them would be very positive and would help many of them to survive, in addition to strengthening their personal coping tools, which were accompanied by both negative and positive feelings.

## Figures and Tables

**Table 1 ijerph-18-08306-t001:** Sociodemographic data of the sample.

Participants	Age (Years)	Duration of GBV (Years)	Time without Contact with Aggressor (Years)
**1**	37	14	3
**2**	31	2	3
**3**	49	18–20	3
**4**	30	5	3
**5**	36	Throughout coexistence	13
**6**	35	9 months	3
**9**	27	2.5	3
**11**	56	30	10
**12**	38	13	3
**13**	53	Since childhood	2
**14**	36	3	3
**18**	62	21	17
**19**	23	4	3
**20**	21	18 months	4
**21**	21	18 months	3
**22**	26	6 months	3 years and 5 months

**Table 2 ijerph-18-08306-t002:** Interviews and narrations.

Data Collection	Questions
Semi-structured interviews and questionnaires	When did you realize you were a victim of gender-based violence? How did you recognize it? How did you feel when you identified it? What did you do to stop the abuse by your ex-partner? What triggered your decision to get out of that situation? How did you feel? What personal and psychological coping tools and mechanisms have you used throughout the process? What tools have you acquired? What social resources and tools did you have available? How did you gain access to them? What difficulties did you find to getting access? Which do you consider were most and least useful? Were any social resources missing?

**Table 3 ijerph-18-08306-t003:** Results.

Topic: Process of Violence	
Subtopics	Relevant Issues of Subtopics
Types of aggressions suffered by the women Identification of violence and actions to stop it Consequences of abuse	Physical and sexual aggression Psychological harm Social destruction Financial aggression Refusing to recognize themselves as victims Fear of physical assault Suffering of children Reporting the violence to the police, leaving the relationship and/or home and calling for help Onset of unspecific diseases Sleep disorders Anxiety Nightmares Hypersensitivity to social image of abused women Inability to visualize the future Not overcoming the process
Topic: Social resources for coping with and overcoming GBV	
Subtopics	Relevant issues of subtopics
Natural social support network Organized social support network	Psychological and physical support to favor empowerment Feeling alone and not helped by the network Did not talk about their experience Their support network would support the aggressor They felt too pressured Feeling accompanied, not judged and not guilty of what happened Dissatisfaction with resources
Topic: Personal tools for coping with and overcoming GBV	
Subtopics	Relevant issues of subtopics
Their skills as the best tool	The process improved their social tools. Capable of accepting and facing all adversities Improvement of their self-esteem Independence from their aggressor
Topic: Feelings identified from the situation of abuse to survival	
Subtopics	Relevant issues of subtopics
When my life was all darkness My world is beginning to shine	Feelings of disgust towards abuse and towards themselves Insecurity and low self-esteem Guilt, shame, fear, anxiety, rage, negation of self-worth Feelings of calm, relief, struggle, indifference, fight, independence, happiness, strength and overcoming

## Data Availability

Not applicable.

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
