# Peer review of "Personal Tools and Psychosocial Resources of Resilient Gender-Based Violence Women"

_ijerph, 2021, doi:10.3390/ijerph18168306_

Round 1
Reviewer 1 Report
The subject of the paper and the research itself are highly significant and they merit special attention and support. The authors stated clearly the data collection process, and the data itself appears to be well organized. Yet, there is a suspicion of a methodological gap. Although the strategy for data analysis is anchored into the seven steps of Paul F. Colaizzi’s method of descriptive phenomenology, the authors ommitted to state if they followed the bracketing process in the formulation of meaning; something that makes the selection and interpretation of data subject to bias. Nevertheless, the paper makes a strong case for theoretical soundness, and it is worth publishing without any reservation.
Reviewer 2 Report
Nothing to mention